

# Modified Early Warning Score (MEWS) combined with biomarkers in predicting 7-day mortality in traumatic brain injury patients in the emergency department: a retrospective cohort study

Shouzhen Zhu[1,*], Yongqiang Yang[2,*], Boling Long[1], Li Tong[3], Jinhua Shen[1] and Xueqing Zhang[3]

[1] Department of Emergency, Changde Hospital, Xiangya School of Medicine, Central South University (The First People's Hospital of Changde City), Changde, China
[2] Department of Neurosurgery, Changde Hospital, Xiangya School of Medicine, Central South University (The First People's Hospital of Changde City), Changde, China
[3] Department of Nursing, Changde Hospital, Xiangya School of Medicine, Central South University (The First People's Hospital of Changde City), Changde, China
* These authors contributed equally to this work.

Corresponding author
Xueqing Zhang,
382731326@QQ.COM

## ABSTRACT

**Background:** Traumatic brain injury (TBI) is a leading cause of injury-related disability and death globally, which negatively affects individuals, families, and society. Predicting the risk for mortality among TBI patients is crucial in guiding further timely and effective treatment plans. Both the standard risk assessment tools and blood-based biomarkers are helpful in predicting outcomes among TBI patients. However, no studies have compared the predicting performance of the individual and combined indicators from the two major types.

**Aim:** This study aimed to compare the Modified Early Warning Score (MEWS), Red blood cell distribution width (RDW), and creatine in predicting 7-day mortality among TBI patients.

**Methods:** A retrospective study was conducted in the emergency department of the First People's Hospital of Changde, China, from January 1, 2023, to June 30, 2023. Data of 1,701 patients with TBI were obtained from the hospital's electronic medical records. A logistic regression model was used to determine independent factors influencing 7-day mortality. The area under the curve (AUC) of the receiver operating characteristic curve (ROC) was calculated to compare the individual and combined effects of MEWS, RDW, and creatine in predicting 7-day mortality based on bootstrap resampling (500 times).

**Results:** Among the 1,701 patients, 225 died, with a mortality rate of 13.23%. The multivariate analysis showed that the type of TBI lesion, MEWS, SBP, DBP, MAP, $SpO_2$, temperature, RDW, and creatine were significantly associated with 7-day mortality. MEWS (AUC: 0.843) performed better than RDW (AUC: 0.785) and creatine (AUC: 0.797) in predicting 7-day mortality. MEWS+RDW (AUC: 0.898) performed better than MEWS+creatine (AUC: 0.875) and RDW+ creatine (AUC: 0.822) in predicting 7-day mortality. The combination of all three indicators, MEWS+RDW+creatine, showed the best predicting performance (AUC: 0.906).

**Conclusion:** MEWS performed best in predicting the 7-day mortality of TBI patients, and its predicting performance was improved when combined with blood-based biomarkers such as RDW and creatine. Our findings provide preliminary evidence supporting the combination of MEWS with blood-based biomarkers as a new method for predicting 7-day mortality in patients with TBI.

# INTRODUCTION

Traumatic brain injury (TBI) is among the top three causes of injury-related disability and death globally, affecting 50 million–60 million people and costing around US$400 billion each year (*Maas et al., 2022*). TBI refers to parenchymal brain damage caused by the application of an external source, manifested as headache, coma, dizziness, confusion, intermittent memory loss, and seizure (*Ritter, 2023*). The main drivers of TBI are road traffic accidents in low-income and middle-income countries (LMICs) and falls among older adults in high-income countries (HICs) (*Maas et al., 2022*). According to the Glasgow Coma Scale (GCS) score, TBI was further graded into mild (GCS 13–15), moderate (GCS 9–13), and severe (GCS 3–8) levels (*Tenovuo et al., 2021*). Mild TBI accounts for over 90% of all TBI cases and nearly half of those mild cases cannot recover to the normal health status before the attack (*Maas et al., 2022*). Severe TBI is associated with an increased risk of brain damage such as swelling, hypotension, and hypoxemia, which, if not treated appropriately, may lead to death (*Hawryluk et al., 2020*). TBI not only causes adverse health outcomes in individuals but also places a substantial burden on the family and society (*Mahajan, Prabhakar & Bilotta, 2023*; *Michael et al., 2023*). Studies in European countries showed that fatal TBIs accounted for 71% of the total TBI disability-adjusted life years (DALYs) (*Te Ao et al., 2015*), and each TBI death was associated with about 24 years of lost life (YLLs) (*Majdan et al., 2017*).

Given the significant disease burden of TBI, it is crucial to conduct accurate mortality risk assessment among TBI patients to guide timely and effective triage to a trauma center or intensive care unit for rescue and treatment (*Najafi, Zakeri & Mirhaghi, 2018*). A timely and accurate pre-assessment of mortality risk among TBI patients can help inform evidence-based clinical decision-making related to the allocation of resources, treatment plans, and further transfer. In order to improve the efficiency and accuracy of mortality prediction with TBI patients at the emergency department, the risk prediction tool should be simple to use, cost-effective, and with high sensitivity and specificity to accurately distinguish between patients with severe and mild injuries (*Gupta et al., 2024*). Several scoring systems, such as the Glasgow Coma Scale (GCS) (*Samuel et al., 2023*), the Revised Trauma Score (RTS) (*Lee et al., 2023*), the quick Sequential Organ Failure Assessment Score (qSOFA) (*Sadhwani, Ambore & Bakhshi, 2022*), and the Modified Early Warning Score (MEWS) (*Wu et al., 2021*), have been widely used as risk prediction tools for patients with TBI in clinical practice. Among these tools, MEWS has demonstrated the best

performance in predicting the risk of mortality. A recent review reported that MEWS could predict multiple adverse events, including transferring to the intensive care unit (ICU) and early mortality in patients with TBI (*Martin-Rodriguez et al., 2020*). A study in Korea demonstrated that MEWS was more accurate in predicting patient admission to the ICU than qSOFA (*Ko et al., 2022*). Another study in China showed that the MEWS had better predictive efficacy than the revised trauma score (RTS) (*Yu, Xu & Chen, 2021*).

Recently, blood-based biomarker measurement has also been shown to be a safe and accessible method to assess the pathologic processes and outcomes of TBI (*Edwards et al., 2023*). For instance, an increase in the blood biomarkers in patients with normal CT results may indicate structural brain damage, which can be confirmed by later MR scanning (*Maas et al., 2022*). One potential biomarker that may be used to predict the mortality risk of TBI is the red blood cell distribution width (RDW). As an indicator of inflammation, RDW plays an essential role in the diagnosis and classification of anemia and has been used in various clinical settings, including cardiac surgery (*Frentiu et al., 2023*), Cholecystitis (*Dong et al., 2021*), type 2 diabetes and foot ulcers (*Hong et al., 2022*), appendicitis (*Anand et al., 2022*), COVID-19 (*Guani-Guerra et al., 2022*), ventilator-associated pneumonia (VAP) (*Nan et al., 2024*), and cancer (*Nocini et al., 2023*). RDW has also been shown to predict various prognosis outcomes among patients with TBI (*Lorente et al., 2021*; *Wang et al., 2020*; *Weihs et al., 2023*). In addition, creatine is another key plasma marker of TBI that can predict intracranial injuries (*Forouzan et al., 2023*). Creatine is an intracellular protein in the central nervous system (CNS) that can provide energy for cells by catalyzing creatine phosphorylation to phosphocreatine (*Gan et al., 2019*). Creatine can prevent and reduce the oxidative stress and damage induced by TBI (*Saraiva et al., 2012*). Furthermore, creatine supplementation has been demonstrated to be effective in preventing TBI and related complications among various populations (*Ainsley Dean et al., 2017*; *Dolan, Gualano & Rawson, 2019*; *Freire Royes & Cassol, 2016*; *Sakellaris et al., 2006*).

Both the standard risk assessment tools and blood-based biomarkers are beneficial in predicting outcomes among TBI patients. One study suggests that a more mature early warning system combining risk assessment tools and biomarkers should be adopted in the future (*Yu, Xu & Chen, 2021*). However, no studies have compared the predicting performance of the individual and combined indicators from the two major types. Therefore, we conducted this retrospective cohort study to compare the individual and combined effects of MEWS, RDW, and creatine in predicting 7-day mortality of TBI patients.

## METHODS

### Study design, participants, and procedure

A retrospective study was conducted among patients with TBI admitted to the emergency department of the First People's Hospital of Changde, China, from January 1, 2023, to June 30, 2023. The inclusion criteria of the participants were as follows: (1) with a confirmed diagnosis of TBI, (2) age ≥18, and (3) with a head Abbreviated Injury Scale (AIS) score ≥3 (*Mellick, Gerhart & Whiteneck, 2003*). The exclusion criteria were as follows: (1) patients
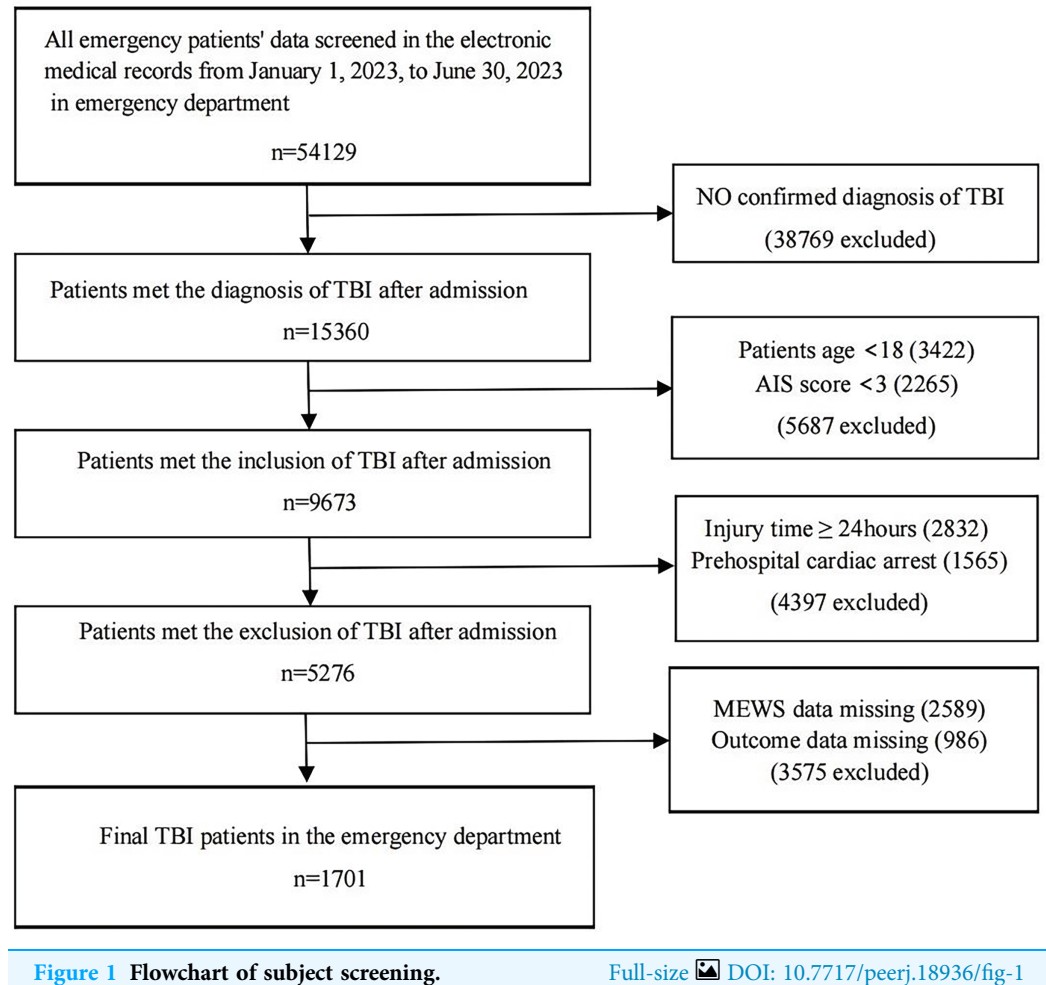

**Figure 1 Flowchart of subject screening.**

who were admitted after 24 h of injury, (2) patients who had a cardiac arrest following trauma before admission, and (3) patients with incomplete information on the predicting and outcome variables. A total of 54,129 patients were admitted to the emergency department during the study period, among whom 15,360 had a confirmed diagnosis of TBI. After excluding 5,687 patients who were aged <18 or had AIS scores <3, a total of 9,673 patients were selected as eligible participants for the study. We further excluded 2,832 patients who were admitted after 24 h, 1,565 patients who had a cardiac arrest before admission, and 3,575 patients with incomplete data on MEWS or mortality, leading to a final sample of 1,701 patients included in the analysis. The detailed patient selection process is illustrated in Fig. 1. When the TBI patient was admitted to the emergency room, the first step was pre-triage, and then the TBI patient was admitted to the monitoring room of the trauma center *via* green passage as the second step. The third step involved the initial control of asphyxia, shock, and massive bleeding before further evaluation. The fourth step included routine tests according to the guidelines, such as white blood cells, red blood cells, and creatine. These tests typically took less than 1 h from sample collection to result reporting. Finally, the definitive treatment was applied to the patient, such as surgery or

**Table 1 Modified Early Warning Score (MEWS).**

| Score | 0 | 1 | 2 | 3 |
|---|---|---|---|---|
| Respiratory rate (min⁻¹) | 9–14 | 15–20 | 21–29/≤8 | ≥30 |
| Heart rate (min⁻¹) | 51–100 | 101–110/41–50 | 111–129/≤40 | ≥130 |
| Systolic BP (mmHg) | 101–199 | 81–100 | ≥200/71–80 | ≤70 |
| Temperature (°C) | 35.1–38.4 | | ≥38.5/≤35.0 | |
| Neurological condition | Alert | Responding to voice | Responding to pain | Unresponsive |

**Note:**
The total score is the sum of each component.

other medical treatment (*American College of Surgeons, 2015*; *Picetti et al., 2019*; *Silverberg et al., 2020*). This study was approved by the Ethics Committee of the First People's Hospital of Changde City (YX-2024-010-01). The requirement for written informed consent was waived due to the retrospective nature of the study.

## Data collection

Data on the following variables were obtained from the electronic medical records: age, gender, leading causes of TBI, type of TBI lesion, hours in the emergency department (HER, hours), respiratory rate (RR, cycles per minute), heart rate (HR, beats per minute), systolic blood pressure (SBP, mmHg), diastolic blood pressure (DBP, mmHg), mean arterial pressure (MAP, mmHg), body temperature (T, °C), and oxygen saturation ($SpO_2$, %) on admission. We also collected the first routine test results when the TBI patient was admitted to the emergency department: white blood cell, red blood cell, RDW, platelet, potassium, creatine, and hemoglobin. GCS and MEWS scores were assessed by triage nurses who have received in-hospital education and training in the triage room at the emergency department. The GCS is calculated based on eye-opening, verbal response, and motor, and the total score ranges from 3 to 15 (*Anand, Shahid & Shameel, 2024*). The MEWS is calculated based on the following vital signs: heart rate, blood pressure, respiratory rate, body temperature, and AVPU (Alert, Voice, Pain, Unresponsive). Details of each variable are shown in Table 1. The total MEWS score ranges from 0 to 3, with a higher score indicating a worse state of TBI patients. AVPU is estimated from the GCS reported previously as follows: A = 14–15, V = 9–13, P = 4–8, U = 3 (*Kim et al., 2021*). The primary outcome was 7-day in-hospital mortality.

## Statistical analyses

Missing data were filled in using the multiple imputation method. Continuous data were described using means ± standard deviations (SD) for normally distributed variables and medians with interquartile ranges (IQR) for non-normally distributed variables. Categorical variables were presented using frequencies and percentages (%). A Student's t-test (normal distribution) or Mann-Whitney U test (non-normal distribution) was used to compare continuous variables by 7-day in-hospital mortality. The chi-square test was used to compare categorical variables by 7-day in-hospital mortality. Influencing factors of the 7-day mortality were determined by the logistic regression model, which models the

log odds of an outcome as a linear combination of one or more independent variables. The algorithm for the logistic regression model is as follows: $y = \beta 0 + \beta 1 X1 + \beta 2 X2 + \ldots + \beta m Xm$, where the coefficient (weight) of each X variable represents the change in the log odds of the Y variable (mortality) for a unit change in the X variable. Receiver operating characteristic (ROC) curve analysis was performed to compare the prognostic performances of the individual and combined effects of MEWS, RDW, and creatine in predicting 7-day in-hospital mortality, and their predicting performance was calculated by the area under the curve (AUC). The Delong tests were conducted to compare the ROC CIs of various indicators. Bootstrap resampling was used to validate the IT model based on the TRIPOD guideline. In particular, bootstrap resampling (500 times) was employed to obtain the area under the curve (AUC) and the best cut-off value of individual and combined indicators of MEWS, RDW, and creatine. Statistical analyses were performed using R (http://www.R-project.org), Empower (R) (http://www.empowerstats.com, X&Y Solutions, Inc., Boston, MA, USA), and SPSS V 23.0. $P < 0.05$ (two-sided) was considered statistically significant.

## RESULTS

### Sample characteristics

Table 2 shows the sample characteristics and their comparisons between the survival and death groups. A total of 1,701 patients with TBI were included in the analysis, including 456 women and 1,245 men. Patients were stratified by 7-day mortality, among whom 1,476 TBI patients survived, and 225 TBI patients died, leading to a mortality rate of 13.23%. Vehicle collision (36.51%) accounted for the largest proportion of all TBI causes. The survival group had higher MEWS scores and spent longer hours (HER) in the emergency room than the death group. The two groups also showed statistically significant differences in the following indicators: the type of TBI lesion, HR, RR, SBP, DBP, MAP, $SpO_2$, temperature, RDW, and potassium.

### Univariate and multivariate analysis

Table 3 shows the univariate and multivariate analysis results of the influencing factors of 7-day mortality. The univariate analysis indicated that MEWS score, the type of TBI lesion, HER, HR, RR, SBP, DBP, MAP, $SpO_2$, temperature, RDW, potassium, and creatine were correlated with 7-day mortality. The multivariate analysis identified the following independent influencing factors associated with 7-day mortality: MEWS score (OR: 1.92, 95% CI [1.70–2.18]), the TBI type of Subdural hematoma (OR: 4.23, 95% CI [1.73–10.30]), Intracerebral hematoma (OR: 3.84, 95% CI [1.55–9.52]), Cerebral contusion (OR: 5.32, 95% CI [2.23–12.72]), Hemorrhagic contusion (OR: 3.09, 95% CI [1.17–8.15]), Penetrating TBI (OR: 3.19, 95% CI [1.24–8.20]), Subarachnoid hemorrhage (OR: 5.31, 95% CI [2.15–13.10]), SBP (OR: 1.11, 95% CI [1.01–1.22]), DBP (OR: 1.23, 95% CI [1.02–1.49], MAP (OR: 0.74, 95% CI [0.56–0.98]), $SpO_2$ (OR: 0.97, 95% CI [0.94–0.99]), temperature (OR: 1.18, 95% CI [1.04–1.34]), RDW (OR: 1.31, 95% CI [1.22–1.41]), creatine (OR: 1.34, 95% CI [1.23–1.47]).

Table 2 Baseline characteristics of TBI patients stratified by 7-day mortality.

| | Total (*n* = 1,701) | Survival (*n* = 1,476) | Death (*n* = 225) | *P*-value | *P*-value* |
|---|---|---|---|---|---|
| **Gender, number (%)** | | | | 0.449 | |
| Male | 1,245 (73.19%) | 1,085 (73.51%) | 160 (71.11%) | | |
| Female | 456 (26.81%) | 391 (26.49%) | 65 (28.89%) | | |
| **Age (year), number (%)** | | | | | |
| age >= 18 and age < 20 | 34 (2.00%) | 28 (1.90%) | 6 (2.67%) | 0.447 | |
| age >= 20 and age < 30 | 188 (11.05%) | 167 (11.31%) | 21 (9.33%) | | |
| age >= 30 and age < 40 | 207 (12.17%) | 187 (12.67%) | 20 (8.89%) | | |
| age >= 40 and age < 50 | 336 (19.75%) | 291 (19.72%) | 45 (20.00%) | | |
| age >= 50 and age < 60 | 396 (23.28%) | 335 (22.70%) | 61 (27.11%) | | |
| age >= 60 and age < 70 | 323 (18.99%) | 286 (19.38%) | 37 (16.44%) | | |
| age >= 70 and age < 80 | 149 (8.76%) | 124 (8.40%) | 25 (11.11%) | | |
| age >= 80 and age < 90 | 61 (3.59%) | 52 (3.52%) | 9 (4.00%) | | |
| age >= 90 | 7 (0.41%) | 6 (0.41%) | 1 (0.44%) | | |
| **TBI type, number (%)** | | | | <0.001 | |
| Epidural hematoma | 282 (16.58%) | 272 (18.43%) | 10 (4.44%) | | |
| Subdural hematoma | 279 (16.40%) | 240 (16.26%) | 39 (17.33%) | | |
| Intracerebral hematoma | 250 (14.70%) | 205 (13.89%) | 45 (20.00%) | | |
| Cerebral contusion | 295 (17.34%) | 243 (16.46%) | 52 (23.11%) | | |
| Hemorrhagic contusion | 198 (11.64%) | 176 (11.92%) | 22 (9.78%) | | |
| Penetrating TBI (dura penetrated) | 215 (12.64%) | 189 (12.80%) | 26 (11.56%) | | |
| Subarachnoid hemorrhage | 182 (10.70%) | 151 (10.23%) | 31 (13.78%) | | |
| **The leading cause of TBI, number (%)** | | | | 0.610 | |
| Fall | 330 (19.40%) | 277 (18.77%) | 53 (23.56%) | | |
| Vehicle collision | 621 (36.51%) | 545 (36.92%) | 76 (33.78%) | | |
| Sports or recreation | 258 (15.17%) | 226 (15.31%) | 32 (14.22%) | | |
| Assault or gunshot | 37 (2.18%) | 33 (2.24%) | 4 (1.78%) | | |
| Hit by object | 207 (12.17%) | 182 (12.33%) | 25 (11.11%) | | |
| Other | 248 (14.58%) | 213 (14.43%) | 35 (15.56%) | | |
| **HER (hour)** | 5.00 (2.00–15.00) | 4.00 (2.00–15.00) | 7.00 (2.00–22.00) | | <0.001 |
| **MEWS score** | 2.00 (1.00–3.00) | 2.00 (1.00–3.00) | 5.00 (3.00–7.00) | | <0.001 |
| **HR (beats/minute)** | 85.00 (75.00–100.00) | 85.00 (75.00–97.00) | 101.00 (78.00–123.00) | | <0.001 |
| **RR (cycles/minute)** | 20.00 (17.00–21.00) | 19.00 (17.00–20.00) | 20.00 (18.00–27.00) | | <0.001 |
| **SBP (mmHg)** | 134.08 ± 30.18 | 135.64 ± 26.20 | 123.80 ± 47.65 | <0.001 | |
| **DBP (mmHg)** | 81.39 ± 19.02 | 82.04 ± 16.25 | 77.19 ± 31.42 | <0.001 | |
| **MAP (mmHg)** | 98.89 ± 20.98 | 99.90 ± 17.86 | 92.21 ± 34.47 | <0.001 | |
| **SpO$_2$ (%)** | 95.59 ± 9.69 | 96.89 ± 5.09 | 87.03 ± 21.38 | <0.001 | |
| **Temperature (°C)** | 36.81 ± 1.22 | 36.88 ± 1.17 | 36.37 ± 1.45 | <0.001 | |
| **WBC (10$^9$/L)** | 10.20 (7.70–14.30) | 10.20 (7.75–14.20) | 10.10 (7.40–14.45) | | 0.403 |
| **RBC (10$^{12}$/L)** | 4.04 (3.42–4.52) | 4.04 (3.44–4.54) | 4.06 (3.32–4.48) | | 0.420 |
| **RDW (%)** | 12.65 ± 3.78 | 12.19 ± 3.70 | 15.66 ± 2.75 | <0.001 | |
| **PLT (10$^9$/L)** | 213.00 (162.00–272.00) | 213.0 (164.0–273.0) | 208.0 (159.25–265.00) | | 0.271 |

(Continued)

| | Total (n = 1,701) | Survival (n = 1,476) | Death (n = 225) | P-value | P-value* |
|---|---|---|---|---|---|
| Potassium (mmol/L) | 4.10 ± 0.59 | 4.07 ± 0.53 | 4.34 ± 0.85 | <0.001 | |
| Creatine (mg/dL) | 1.00 (0.80–1.50) | 1.00 (0.80–1.40) | 1.40 (0.80–5.60) | | <0.001 |
| Hemoglobin (g/L) | 118.6 ± 24.30 | 118.9 ± 24.1 | 116.6 ± 24.9 | 0.345 | |

Notes:

TBI, Traumatic Brain Injury; HER, Hours In the Emergency Room; MEWS, Modified Early Warning Score; HR, Heart rate; RR, Respiratory Rate; SBP, Systolic Blood Pressure; DBP, Diastolic Blood Pressure; MAP, Mean Artery Pressure; $SPO_2$, Percutaneous Oxygen Saturation; °C, Celsius; %, Percent; WBC, White Blood Cell; RBC, Red Blood Cell; RDW, Red Cell Distribution Width; PLT, Platelet.

P-value: Student's t-test for continuous variables was normal distribution, Chi-square test for categorical variable; P-value*: Mann-Whitney U-tests for continuous variables with non-normal distribution.

**Table 3 Analysis of influencing factors of 7-day mortality in TBI patients.**

| Variable | Univariate | | Multivariate | |
|---|---|---|---|---|
| | OR (95% CI) | P-value | OR (95% CI) | P-value |
| **Gender, n (%)** | | | | |
| Male | Reference | | | |
| Female | 1.13 [0.83–1.54] | 0.4495 | | |
| **Age (year), number (%)** | | | | |
| age ≥ 18 and age < 20 | Reference | | | |
| age ≥ 20 and age < 30 | 0.59 [0.22–1.58] | 0.2921 | | |
| age ≥ 30 and age < 40 | 0.50 [0.18–1.35] | 0.1710 | | |
| age ≥ 40 and age < 50 | 0.72 [0.28–1.84] | 0.4945 | | |
| age ≥ 50 and age < 60 | 0.85 [0.34–2.14] | 0.7295 | | |
| age ≥ 60 and age < 70 | 0.60 [0.23–1.55] | 0.2957 | | |
| age ≥ 70 and age < 80 | 0.94 [0.35–2.51] | 0.9030 | | |
| age ≥ 80 and age < 90 | 0.81 (0.26–2.50] | 0.7112 | | |
| age >= 90 | 0.78 [0.08–7.71] | 0.8299 | | |
| **TBI type, number (%)** | | | | |
| Epidural hematoma | Reference | | Reference | |
| Subdural hematoma | 4.42 [2.16–9.05] | <0.0001*** | 4.23 [1.73–10.30] | 0.0015** |
| Intracerebral hematoma | 5.97 [2.94–12.13] | <0.0001*** | 3.84 [1.55–9.52] | 0.0036** |
| Cerebral contusion | 5.82 [2.89–11.70] | <0.0001*** | 5.32 [2.23–12.72] | 0.0002*** |
| Hemorrhagic contusion | 3.40 [1.57–7.35] | 0.0019** | 3.09 [1.17–8.15] | 0.0227* |
| Penetrating TBI (dura penetrated) | 3.74 [1.76–7.94] | 0.0006*** | 3.19 [1.24–8.20] | 0.0162* |
| Subarachnoid hemorrhage | 5.58 [2.66–11.70] | <0.0001*** | 5.31 [2.15–13.10] | 0.0003*** |
| **The leading causes of TBI** | | | | |
| Fall | Reference | | | |
| Motor vehicle collision | 0.73 [0.50–1.07] | 0.1022 | | |
| Sports or recreation | 0.74 [0.46–1.19] | 0.2118 | | |
| Assault or gunshot | 0.63 [0.22–1.86] | 0.4068 | | |
| Hit by object | 0.72 [0.43–1.20] | 0.2037 | | |
| Other | 0.86 [0.54–1.36] | 0.5191 | | |

| Table 3 (continued) | | | | |
|---|---|---|---|---|
| Variable | Univariate | | Multivariate | |
| | OR (95% CI) | *P*-value | OR (95% CI) | *P*-value |
| HER (hour) | 1.01 [1.01–1.03] | 0.0025** | 1.00 [1.00–1.01] | 0.0866 |
| MEWS score | 2.04 [1.87–2.23] | <0.0001*** | 1.92 [1.70–2.18] | <0.0001*** |
| HR (times/minute) | 1.02 [1.02–1.03] | <0.0001*** | 1.00 [0.99–1.01] | 0.9649 |
| RR (times/minute) | 1.08 [1.06–1.11] | <0.0001*** | 1.02 [0.98–1.05] | 0.3341 |
| SBP (mmHg) | 0.99 [0.98–0.99] | <0.0001*** | 1.11 [1.01–1.22] | 0.0369* |
| DBP (mmHg) | 0.99 [0.98–0.99] | 0.0004*** | 1.23 [1.02–1.49] | 0.0311* |
| MAP (mmHg) | 0.98 [0.98–0.99] | <0.0001*** | 0.74 [0.56–0.98] | 0.0366* |
| SpO$_2$ (%) | 0.89 [0.87–0.91] | <0.0001*** | 0.97 [0.94–0.99] | 0.0172* |
| Temperature (°C) | 0.65 [0.55–0.76] | <0.0001*** | 1.18 [1.04–1.34] | 0.0093** |
| WBC ($10^9$/L) | 1.00 [0.99–1.02] | 0.5786 | | |
| RBC ($10^{12}$/L) | 0.93 [0.78–1.11] | 0.4020 | | |
| RDW (%) | 1.46 [1.37–1.56] | <0.0001*** | 1.30 [1.20–1.40] | <0.0001*** |
| PLT (109/L) | 1.00 [0.95–1.01] 0.4356 | | | |
| Potassium (mmol/L) | 2.10 [1.66–2.65] | <0.0001*** | 1.09 [0.77–1.55] | 0.6175 |
| Creatine (mg/dL) | 1.50 [1.40–1.61] | <0.0001*** | 1.34 [1.23–1.47] | <0.0001*** |
| Hemoglobin (g/L) | 0.96 [0.91–1.02] | 0.1976 | | |

**Notes:** TBI, Traumatic Brain Injury; HER, Hours In the Emergency Room; MEWS, Modified Early Warning Score; HR, Heart rate; RR, Respiratory Rate; SBP, Systolic Blood Pressure; DBP, Diastolic Blood Pressure; MAP, Mean Artery Pressure; SPO$_2$, Percutaneous Oxygen Saturation; °C, Celsius; %, Percent; WBC, White Blood Cell; RBC, Red Blood Cell; RDW, Red Cell Distribution Width; PLT, Platelet; 95% CI, 95% Confidence Interval.
* $P < 0.05$.
** $P < 0.01$.
*** $P < 0.001$.

## ROC analysis

Figures 2A–2G show the smooth and raw ROC curves of MEWS, RDW, and creatine with and without the bootstrapping method. The AUC for MEWS, creatine, and RDW were 0.843, 0.797, and 0.785, respectively. The Delong tests showed significant differences in the ROC CIs of the three indicators (Table 4). Moreover, the best thresholds of MEWS, creatine, and RDW were 3.5 (sensitivity: 0.7422, specificity: 0.8448), 3.15 (sensitivity: 0.4359, specificity: 0.9385), and 14.05 (sensitivity: 0.6800, specificity: 0.6829), respectively (Table 5). The smooth ROCs obtained by the bootstrapping method (resample: 500) of MEWS, RDW, and creatine in predicting 7-day mortality are included in the Fig. S1.

Figures 3A–3C show the raw and smooth ROC of MEWS combined with RDW and creatine, with an AUC of 0.906. The AUC of the smooth ROC of MEWS combined with RDW (Model 1) was 0.898, the AUC of the smooth ROC of MEWS combined with creatine (Model 2) was 0.875, and the AUC of smooth ROC of RDW combined with creatine (Model 3) was 0.822. The ROC curves of two indicator combinations with and without the bootstrapping method are included in Fig. S2.

## DISCUSSION

TBI is a significant public health challenge and a leading cause of disease burden worldwide. Compared with HICs, LMICs, such as China, are more adversely affected by

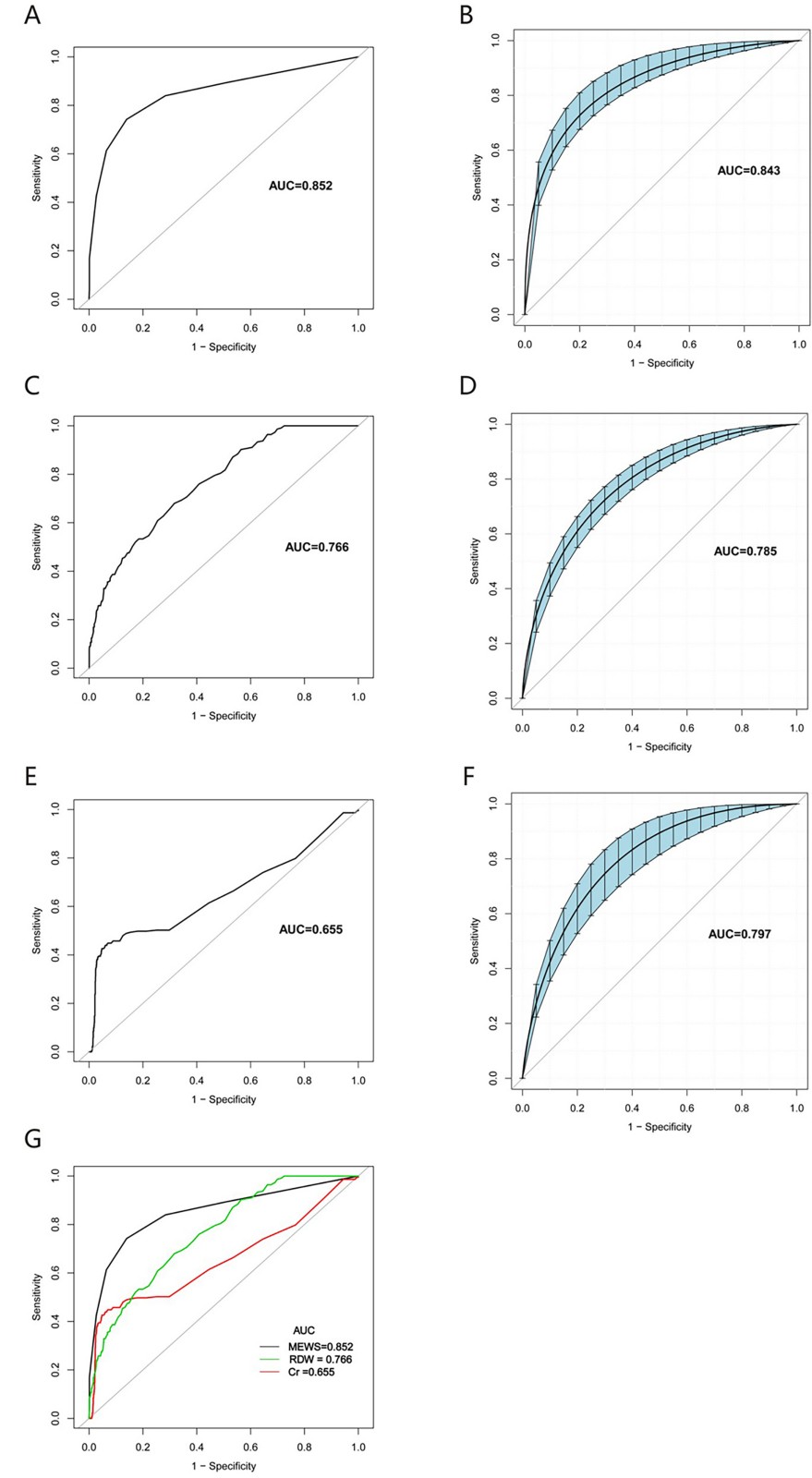

**Figure 2 The smooth and raw ROC curve of MEWS, RDW, and creatine for the prediction of 7-day mortality in TBI patients with and without the bootstrapping method (resample: 500), with an area**

**Figure 2** (continued)

**under the receiver operating characteristic curve.** (A) raw ROC curve of MEWS; (B) smooth ROC curve of MEWS; (C) raw ROC curve of RDW; (D) smooth ROC curve of RDW; (E) raw ROC curve of creatine; (F) smooth ROC curve of creatine; (G) The comparison of raw ROC curves with MEWS, RDW, and creatine.

**Table 4 The AUC of MEWS, RDW, and Creatinine compared by the DeLong test.**

| Variable | AUC (95% CI) | Z-score statistic | *P*-value |
|---|---|---|---|
| MEWS | 0.852 [0.82–0.88] | 4.367 | <0.001 |
| RDW | 0.766 [0.73–0.80] | | |
| MEWS | 0.852 [0.82–0.88] | 7.092 | <0.001 |
| Creatinine | 0.655 [0.61–0.70] | | |
| RDW | 0.766 [0.73–0.80] | 4.076 | <0.001 |
| Creatinine | 0.655 [0.61–0.70] | | |

Notes:
MEWS, modified early warning score; RDW, red cell distribution width; AUC, area under the receiver operating characteristic curve; 95% CI, 95% confidence interval.

**Table 5 Sensitivities, specificities, and predictive values for MEWS with markers in TBI patients.**

| Variable | Best threshold | AUC (95% CI) | Sensitivity% | Specificity% | PPV% | NPV% |
|---|---|---|---|---|---|---|
| **Model with one variable** | | | | | | |
| MEWS | 3.5 | 0.843 [0.81–0.89] | 0.7422 | 0.8448 | 0.4477 | 0.9563 |
| Creatine | 3.15 | 0.797 [0.74–0.84] | 0.4395 | 0.9395 | 0.5241 | 0.9170 |
| RDW | 14.05 | 0.785 [0.75–0.81] | 0.6800 | 0.6829 | 0.2464 | 0.9333 |
| **Model with two variables** | | | | | | |
| Model 1: MEWS+RDW | / | 0.898 [0.88–0.92] | 0.7848 | 0.8830 | 0.5043 | 0.9643 |
| Model 2: MEWS+Creatine | / | 0.875 [0.85–0.90] | 0.8341 | 0.8211 | 0.4143 | 0.9703 |
| Model 3: RDW+Creatine | / | 0.822 [0.79–0.85] | 0.6457 | 0.8544 | 0.4022 | 0.9408 |
| **Model with three variables** | | | | | | |
| MEWS+RDW+Creatine | / | 0.906 [0.88–0.93] | 0.8341 | 0.8823 | 0.5181 | 0.9722 |

Notes:
MEWS, Modified Early Warning Score; RDW, Red Cell Distribution Width; AUC, Area Under Curve; 95% CI, 95% Confidence Interval; PPV, Positive Predictive Value; NPV, Negative Predictive Value.

the disease burden of TBI due to less developed systems, protocols, and programs related to urgent care in the trauma center of the emergency department (*Yu, Xu & Chen, 2021*). An accurate and cost-effective risk-predicting model for TBI is essential to guide effective and targeted treatment plans. The Lancet Neurology Commission on TBI, published in 2022, called for a concerted effort to use blood-based biomarkers, which is a breakthrough for TBI diagnosis and prognosis (*Maas et al., 2022*). In consideration of the traditional trauma scoring systems' limitations and the Lancet Neurology Commission's suggestion on biomarkers, we conducted this study to assess and compare the individual and combined effects of MEWS, RDW, and creatine in predicting the 7-day mortality of TBI patients.

Our results showed a mortality rate of 13.23% among TBI patients, indicating that at least one in ten TBI patients would die and thus warrant close clinical attention. The

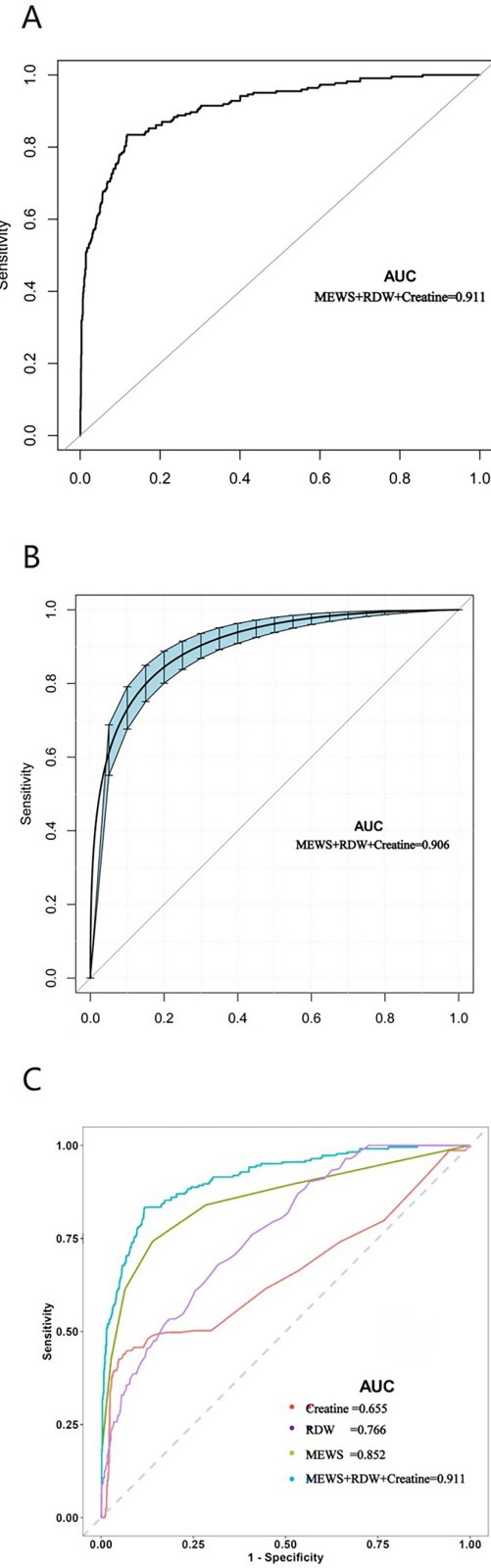

**Figure 3** The ROC curve of the combined effects of MEWS, RDW, and creatine in predicti ng 7-day mortality in TBI patients with and without in the emergency department obtained by the bootstrapping

**Figure 3 (continued)**
**method (resample: 500), with an area under the receiver operating characteristic curve.** (A) Raw ROC curve of MEWS combined RDW, and creatine; (B) smooth ROC curve of MEWS combined RDW, and creatine; (C) the comparison of raw ROC curves between three indictors combination with single indictor.

leading causes of TBI were falls and vehicle collisions, which is consistent with previous studies found in developing countries. Both univariate and multivariate analyses showed that the TBI type was associated with 7-day mortality, which is also congruent with previous findings (*Abujaber et al., 2020*; *Whitehouse et al., 2022*). The type of TBI is determined based on the location of intracranial bleeding, and each location has its unique characteristics and susceptibility to damages, leading to different mortality risks. In addition, MEWS, SBP, DBP, MAP, SpO$_2$, temperature, RDW, and creatine were independent influencing factors associated with 7-day mortality. Furthermore, the comparison of single indicators showed that MEWS outperformed RDW and creatine in predicting 7-day mortality among TBI patients. The comparison of the two indicator combinations showed that MEWS+RDW performed better than MEWS+creatine and RDW+creatine. Finally, the combination of all three indicators, MEWS+RDW+creatine, showed the best predicting performance.

To our knowledge, no other study has combined traditional risk assessment tools with blood-based biomarkers to predict the prognosis of TBI patients in the emergency department. Our study represents the first effort in this area by comparing the individual and combined predictive values of MEWS, creatine, and RDW in 7-day mortality among TBI patients in the emergency department. Our results showed that the AUC of MEWS was statistically higher than that of creatine and RDW, indicating that MEWS performed superior to creatine and RDW in predicting 7-day mortality. Furthermore, the combination of MEWS+ RDW performed better than the other two indicator combinations, and the combination of all three indicators showed the best predicting performance. Both RDW and creatine are easy to obtain in the emergency department, and this study provides a new direction in using MEWS combined with biomarkers to predict 7-day mortality with TBI patients in the emergency department.

The MEWS includes the GCS score and vital signs, which have been shown to perform better in predicting mortality than the RTS, ISS, and SI in TBI patients (*Kim et al., 2021*; *Yu, Xu & Chen, 2021*). A previous study showed that higher MEWS was related to injury severity, mortality, and intensive care unit (ICU) admission and could also be used for predicting massive transfusion (MT) in severe trauma patients with TBI (*Chae et al., 2022*). In our study, the best threshold of MEWS was 3.5, consistent with a recent study, which showed that MEWS ≥ 3 calculated 12–36 h before positive blood culture was a predictor of ICU admission (*Allarakia et al., 2023*). RDW is an important indicator of inflammation and has been applied to various clinical conditions to predict prognosis. In our study, RDW also showed good predicting performance in 7-day mortality with TBI patients. This finding was consistent with studies conducted in China and Austria (*Wang et al., 2020*; *Weihs et al., 2023*), indicating that RDW may be a useful tool for estimating the prognosis of TBI patients (*Lorente et al., 2021*). In addition, our study also confirmed the predicting
effect of creatine in 7-day mortality with TBI patients, which was consistent with previous studies showing similar results (*Ainsley Dean et al., 2017*; *Dolan, Gualano & Rawson, 2019*; *Freire Royes & Cassol, 2016*; *Sakellaris et al., 2006*). One study containing 421 TBI patients concluded that creatine combined with other prognostic factors could help to predict the outcomes in male TBI patients (*Zheng et al., 2021*). Another longitudinal biochemical profiling of TBI reported that the concentrations of taurine, creatine, adenine, dimethylamine, histidine, N-Acetyl aspartate, and glucose 1-phosphate were all associated with TBI severity (*Yilmaz et al., 2023*). TBI patients with chronic kidney disease tend to have a poorer prognosis than those with normal kidney function (*Mo et al., 2023*). Our study identified creatine as a serum biomarker that can be used to predict adverse outcomes in TBI events.

## Strengths

A major innovation of the study was that we presented a new concept algorithm to predict 7-day in-hospital mortality based on the type of TBI lesions, physical examination test (such as blood pressure, body temperature, and oxygen saturation), and standard laboratory test (such as MEWS, RDW, and creatine). All these parameters are easily accessible as part of the routine test for TBI patients admitted to the hospital. Therefore, our risk prediction model provides a relatively rapid, reliable, and cost-effective way to identify high-risk TBI patients and take proactive interventions to prevent and reduce mortality. Other advantages of the study included a large sample size to ensure enough statistical power to detect significant results, the inclusion of a variety of clinical indicators to control for confounding effects, and a comprehensive comparison of the predictive performance of the individual and various combinations of the indicators.

## Clinical limitations and future directions

However, this study also had several limitations. First, the retrospective nature of the study design is susceptible to recall bias and missing information. Future prospective study designs are needed to produce more robust results. Second, all participants were recruited from a single hospital and may not represent TBI patients in other hospitals and other regions. Future multicenter studies are needed to validate our model further. Third, we did not analyze the effects of essential procedures (such as interventions, operations, and transfusions) on 7-day mortality. Future studies should consider adding these variables to control for their potential confounding effects. Fourth, we only focused on three biomarkers based on laboratory tests and did not include other biomarkers, such as genetic markers, which can provide more specific information on the mechanism of TBI-related death. Future studies may consider adding genetic markers, such as S100B, glial fibrillary acidic protein (GFAP), ubiquitin C-terminal hydrolase-L1 (UCH-L1), and neurofilament light (NF-L), to get a deeper understanding of the TBI mechanism. Finally, it should be noted that although the three biomarkers are easily available as part of the routine test for TBI patients admitted to the emergency room, they are not as quickly available as MEWS, which takes only a few minutes rather than up to 1 h. Although the combination of MEWS and biomarkers may improve the diagnostic performance in predicting 7-day mortality, its

clinical utility in time-sensitive emergency care settings may be limited. Therefore, in real practice, clinicians should choose the most appropriate indicators by balancing the prediction accuracy and the time cost. Future studies should explore other time-saving and cost-effective risk prediction models to predict mortality among patients with TBI admitted to the emergency department.

### Implications

In this study, we developed a 7-day in-hospital risk prediction model based on readily available parameters, such as MEWS, RDW, and creatine, which carry significant implications in clinical practice. Compared to the expensive and time-consuming genetic tests such as S100B and GFAP, our model offers a more affordable and practical evaluation of mortality risk among TBI patients, which is especially valuable in resource-limited rural areas and less developed countries. The finding that MEWS outperformed RDW and creatine in predicting 7-day mortality and that the combination of all three indicators had the best-predicting performance also provides helpful, practical guidance to inform clinical decision-making. As soon as the TBI patient is admitted to the emergency department, the more readily available indicator, MEWS, can be used to provide a quick and reliable prediction. After the doctor confirms the first diagnosis, the combination of MEWS with RDW and creatine may be considered to get a more precise prediction.

## CONCLUSION

MEWS performed best in predicting the 7-day mortality of TBI patients, and its predicting performance was improved when combined with blood-based biomarkers such as RDW and creatine. Our findings provide preliminary evidence supporting the combination of MEWS with blood-based biomarkers as a new method for predicting 7-day mortality in patients with TBI.

## ACKNOWLEDGEMENTS

We thank the participants, patients, and investigators associated with the study.

### Funding

The study was supported by the Hunan Science and Technology Department-Clinical Medical Technology Demonstration Base for Neurosurgery in Hunan Province (2017SK51304). The funders had no role in study design, data collection and analysis, decision to publish, or preparation of the manuscript.

### Grant Disclosures

The following grant information was disclosed by the authors:
Hunan Science and Technology Department-Clinical Medical Technology Demonstration Base for Neurosurgery in Hunan Province: 2017SK51304.

## Competing Interests

The authors declare that they have no competing interests.

## Author Contributions

- Shouzhen Zhu performed the experiments, analyzed the data, prepared figures and/or tables, authored or reviewed drafts of the article, and approved the final draft.
- Yongqiang Yang performed the experiments, authored or reviewed drafts of the article, and approved the final draft.
- Boling Long performed the experiments, authored or reviewed drafts of the article, and approved the final draft.
- Li Tong performed the experiments, authored or reviewed drafts of the article, and approved the final draft.
- Jinhua Shen performed the experiments, prepared figures and/or tables, authored or reviewed drafts of the article, and approved the final draft.
- Xueqing Zhang conceived and designed the experiments, analyzed the data, prepared figures and/or tables, authored or reviewed drafts of the article, and approved the final draft.

## Human Ethics

The following information was supplied relating to ethical approvals (*i.e.*, approving body and any reference numbers):

The Ethics Committee of the First People's Hospital of Changde approved the study

## Data Availability

The raw measurements are available in the Supplemental Files.

## Supplemental Information

Supplemental information for this article can be found online at http://dx.doi.org/10.7717/peerj.18936#supplemental-information.

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
