# Peer review of "Modified Early Warning Score (MEWS) combined with biomarkers in predicting 7-day mortality in traumatic brain injury patients in the emergency department: a retrospective cohort study"

_PeerJ, doi:10.7717/peerj.18936_

## Round 0.1 · original submission · Major Revisions

Both reviewers have expressed serious concerns which must be thoroughly addressed in any revision

·

Basic reporting

This is an interesting study to predict the mortality rate in Traumatic Brain Injury (TBI). However the type of TBI lesion did not presented on this study. There are many factors NOT only MEWS but also the lesion intracranial itself. The TBI can also be classified into focal injuries-including penetrating trauma, cortical/white matter contusion, epi-and subural hematomas-or diffuse injury with wide-spread damage to the cerebrovascular system and/or the white matter. These may influence the outcomes. Biomarkers of TBI could be proteins, metabolites, or other substances such genetic markers that give more specific in TBI mechanism related with outcomes. The mostly studied biomarkers are S100B, Glial fibrillary acidic protein (GFAP), Ubiquitin C-terminal hydrolase-L1 (UCH-L1), Neurofilament light (NF-L). So it would be good if these parameter are also available. The table 3 presentation will be better if the P value is in different collumn and marked if it is significant value.

Experimental design

The prospective study is better to predict the mortality rate within 7 days. If the authors will analyze the retrospective data, it would be good if the authors present the algorithm how to predict the outcomes so that it will be easier to understand by the audience. And I think this is not a specific biomarkers design like S100B, Glial fibrillary acidic protein (GFAP), Ubiquitin C-terminal hydrolase-L1 (UCH-L1), Neurofilament light (NF-L). but it is laboratory test standard like creatinine and so on.

Validity of the findings

It should be clearly define to get novelty of this study, by presenting the new concept algorithm of 7 days prediction of mortality based on standard laboratory test in relation with type of lession intracranially. Because each location of intracranial bleeding have their own characteristics.

Additional comments

It would be very useful study for rural hospital in case such an advanced biomarkers like S100B, Glial fibrillary acidic protein (GFAP), Ubiquitin C-terminal hydrolase-L1 (UCH-L1), Neurofilament light (NF-L) are not available.

Reviewer 2 ·

Basic reporting

This study investigated the prediction effect of MEWS combined with blood based biomarkers on mortality of TBI patients. The story and clinical significance are not clear. The results are not convincible. The authors should also state more details of data inclusion. Furthermore, there are several grammatical mistakes and typos in figure captions. The manuscript should be edited by an English native speaker before re-submission.

Experimental design

The authors stated that A timely and accurate pre-assessment of mortality risk among TBI patients can help inform evidence-based clinical decision-making related to the allocation of resources, treatment plans, and further transfer.
However, only MEWS is simple to use and cost-effective.
RDW and Creatinine are time consuming and requires special devices to measure.
Hence, RDW and Creatinine should not be considered used with MEWS and even be used for the timely diagnosis in emergency department. It should be used for a precise diagnosis after the first diagnosis from doctors.

In methods, there are no description of inclusion and exclusion of the cohort.
There is also no information about how authors deal with the missing values.
The authors should read more other papers to enhance the description of this section.

Validity of the findings

From the ROC results, I find there are overlaps between 95 CIs.
Hence, current ROC results cannot lead to the conclusions.
The authors should conduct the Delong tests to compare ROC CIs.

Also, the authors should plot ROC curves on one figure to compare instead of plotting each of them on a new figure. The ROC curve shape also seems to be manipulated. Usually, the curve should be a zig-zag like curve. Current plot is too smooth.

The authors should also discuss the time-cost and resource of cost for obtaining the values biomarkers. Personally, I think they are much more time consuming and device dependent compared to triage tools like MEWS.

---

## Round 0.2 · Major Revisions

Reviewer 2 still has several concerns which must be addressed appropriately.

·

Basic reporting

The authors have all revised as I suggested to them. I think literaturly has been described clearly and coud be accepted for me.

Experimental design

The research design also well revised by authors as well as methods and analysis.

Validity of the findings

Additional data and information have been well defined and described in the figure and tables.

Additional comments

It can be accepted for publication.

Reviewer 2 ·

Basic reporting

Comment 2:
The authors should further describe clearly the procedures of patient admission and treatment.
How many steps to admit the patient. Which step is routine test?
The authors should also report the approximate time cost (specific quantitative value) for routine test to show its efficiency.

>>Compared to the expensive and time-consuming genetic tests such as S100B and GFAP
such comparison is meaningless.
If genetic tests are common tests in the emergency room, such comparison is meaningful, but I dont think it is the fact.

Furthermore, after admission, to predict the mortality of patient, will you do routine test every day to calculate? That will create burden on patients.

Please reconsider your research story and clinical significance.

Comment 3:
OK, but please also add the figure like other papers.
https://www.researchgate.net/figure/Flow-chart-for-population-inclusion-criteria_fig1_369990787

Comment 4:
OK

Comment 5:
bootstrap resampling is just a data process method.
I was asking you to plot the data process results in one figure to compare...
You can output the results to a csv file, and then plot them in one figure by Excel.
Or you can use R package to plot.

Comment 6:
Same as Comment 2.

Experimental design

N/A

Validity of the findings

N/A

Additional comments

N/A

---

## Round 0.3 · Major Revisions

Please address these final comments from the reviewer

Reviewer 2 ·

Basic reporting

Comment 1:
As i mentioned in first round revision, please discuss why combination of MEWS and bio-marker is reasonable from the viewpoint of time cost.
MEWS can be measured in several minutes.
However, as you described, your measurement of bio-marker required one hour.
Why a patient is needed to wait for the result of bio-marker tests?
Anyhow, the conclusion is meaningful for a general department but not useful in the emergency room.
The authors should discuss it.

Comment 3:
OK

Comment 5:
The revision is not reflected in the manuscript.
Comparison is the more important. Put 2a-2c and 3a-3d to supplementary files.
Remove fig. 2a-2c, and then create the comparison figure with all RoC curves like S1-D1.
Also, remove fig. 3a-3d, and then create the comparison figure with all RoC curves.

Experimental design

NA

Validity of the findings

NA

Additional comments

NA

---

## Round 0.4 · Minor Revisions

Please put Fig. S1-d1 and Fig. S4-d3 in the manuscript instead of supplementary files as suggested by the reviewer. Also authors are suggested to add clinical limitations of this study along with the future directions.

Reviewer 2 ·

Basic reporting

All problems have been addressed.
Please put Fig. S1-d1 and Fig. S4-d3 in the manuscript instead of supplementary files.

Experimental design

NA

Validity of the findings

NA

Additional comments

NA

---

## Round 0.5 · accepted · Accept

Authors have addressed all of the reviewers' comments and manuscript is ready for the publication.